# Temporal Dynamics of Purinergic Receptor Expression in the Lungs of Marek’s Disease (MD) Virus-Infected Chickens Resistant or Susceptible to MD

**DOI:** 10.3390/v16071130

**Published:** 2024-07-14

**Authors:** Haji Akbar, Keith W. Jarosinski

**Affiliations:** Department of Pathobiology, College of Veterinary Medicine, University of Illinois at Urbana-Champaign, Urbana, IL 61802, USA; akbar3@illinois.edu

**Keywords:** Marek’s disease virus (MDV), purinergic receptors, signaling, lung, natural infection

## Abstract

Marek’s disease virus (MDV) is an economic concern for the poultry industry due to its poorly understood pathophysiology. Purinergic receptors (PRs) are potential therapeutic targets for viral infections, including herpesviruses, prompting our investigation into their role in MDV pathogenesis. The current study is part of an experimental series analyzing the expression of PRs during MDV infection. To address the early or short-acting P2 PR responses during natural MDV infection, we performed an “exposure” experiment where age-matched chickens were exposed to experimentally infected shedders to initiate natural infection. In addition, select non-PR regulatory gene responses were measured. Two groups of naïve contact chickens (n = 5/breed/time point) from MD-resistant (White Leghorns: WL) and -susceptible (Pure Columbian) chicken lines were housed separately with experimentally infected PC (×PC) and WL (×WL) chickens for 6 or 24 h. Whole lung lavage cells (WLLC) were collected, RNA was extracted, and RT-qPCR assays were used to measure specific PR responses. In addition, other potentially important markers in pathophysiology were measured. Our study revealed that WL chickens exhibited higher P1 PR expression during natural infection. WL chickens also showed higher expression of *P1A3* and *P2X3* at 6 and 24 h when exposed to PC-infected chickens. *P2X5* and *P2Y1* showed higher expression at 6 h, while P2Y5 showed higher expression at 6 and 24 h; regardless of the chicken line, PC chickens exhibited higher expression of *P2X2, P2Y8*, *P2Y10*, *P2Y13*, and *P2Y14* when exposed to either group of infected chickens. In addition, MDV infection altered the expression of *DDX5* in both WL and PC groups exposed to PC-infected birds only. However, irrespective of the source of exposure, BCL2 and *ANGPTL4* showed higher expression in both WL and PC. The expression of *STAT1A* and *STAT5A* was influenced by time and breed, with major changes observed in *STAT5A*. *CAT* and *SOD1* expression significantly increased in both WL and PC birds, regardless of the source of infection. *GPX1* and *GPX2* expression also increased in both WL and PC, although overall lower expression was observed in PC chickens at 24 h compared to 6 h. Our data suggest systemic changes in the host during early infection, indicated by the altered expression of PRs, *DDX5*, *BCL2*, *ANGPTL4*, and other regulatory genes during early MDV infection. The relative expression of these responses in PC and WL chickens suggests they may play a key role in their response to natural MDV infection in the lungs and long-term pathogenesis and survival.

## 1. Introduction

Marek’s disease (MD) is a highly contagious disease of chickens, caused by gallid alphaherpesvirus 2, species *Mardivirus gallidalpha2*, more widely known as Marek’s disease virus (MDV). Since 1969, efforts have been made to control MD through the widespread use of vaccination that reduces viral replication and lymphoma formation in chickens but does not halt viral shedding and spread within a flock [1]. MDV infection begins through the respiratory route, either through direct bird-to-bird contact or indirectly through inhaling dust and dander from infected birds [2]. The primary cells targeted by MDV infection are associated with the host immune system, including B and T lymphocytes and macrophages that transport the virus systemically to lymphoid organs and the skin, with the latter being the site of release from chickens.

ATP and its metabolites (ADP, cAMP, adenosine) are primarily considered energy sources and nucleic acid building blocks; however, they also play a significant role as extracellular signaling molecules (eATP) in physiological and pathological conditions. Infected or damaged cells release eATP as a “find me signal” into the extracellular environment by acting as an autocrine and paracrine mediator, initiating responses against bacterial, fungal, or viral infections [3,4]. These purines bind to purinergic receptors (PRs), a diverse family of membrane-bound receptors important in numerous biological processes like neuromodulation, inflammation, endothelial-mediated vasodilatation, cell migration, cell proliferation, differentiation, and apoptosis [3,5]. PRs are classified into P1 (adenosine receptors) and P2 (ATP/UTP receptors) subfamilies, further divided into specific receptor types. P1 receptors comprise four subfamilies (A1, A2A, A2B, and A3), while P2 receptors are further divided into two subfamilies, P2X (P2X: 1–7) and P2Y (P2Y: 1–14) receptors [6,7,8]. Most of these receptors have been identified in chickens, at least at the genomic level [9]. In addition to the above classifications and subdivisions, purinergic responses can be either early (short) or late (long) acting based on the infection and physiological condition of the host [10].

Activation of these receptors can either promote or inhibit infection processes [11]. Studies have shown varying effects of different purinergic receptors on virus replication and immune responses, suggesting potential targets for antiviral therapies—for example, Chen et al. [12], reported that the P2Y2 receptor boosts human cytomegalovirus (HCMV) replication, while P2X5 inhibits virus replication [12]. Similar effects have been observed elsewhere; for instance, P2X7R activation enhances the immune killing of tumor cells, whereas A2BR signaling contributes to immunosuppression in tumors [13,14]. Zhang and colleagues [15] also found that P2Y13 could be a potential antiviral target, limiting the replication of various viruses such as Newcastle disease virus (NDV) and herpes simplex virus 1 (HSV1) [15]. Differential expression of PRs has been reported in various pathological conditions [16,17]. Considering this information, along with the clinical aspects of MD and changes in target cells like B and T cells, potential roles for PRs (P1 and P2) in MDV pathogenesis are warranted. Although limited studies have explored purinergic receptor expression in response to herpesvirus infections [6,12,18], our ongoing work extends our previous study of examining alterations in PR expression during MDV infection during infection and induction of disease on different chicken lines in the liver and whole lung lavage cell (WLLC) suspensions [9].

Our recent research has unveiled significant changes in PR expression during MDV infection in chickens [9]. These findings underscore the pivotal role of the host chicken’s genetic background in regulating purinergic receptors, which are intricately linked to disease progression and severity. However, the specific types of PRs expressed during the early stages of MDV infection remain unknown. Hence, this study was designed to test the hypothesis that PR expression is crucial during the early phases of natural MDV infection in chickens, with the host’s genetic background influencing the purinergic response. Our previous report examined the expression of PRs late during MDV infection. Our present study measured the expression of PRs and other potentially important genes’ responses early during natural MDV infection. Moreover, the genetic makeup of viral shedder birds (experimentally infected chickens) may impact the regulation of PRs in new hosts. It has been demonstrated that sentinel contact birds can be efficiently infected when exposed to shedders for as little as 4 h [19]. Hence, we employed a natural virus-host animal model to assess responses 6 and 24 h after exposure to MDV-infected chickens. Since the respiratory tract is the primary route of MDV entry, initially targeting pulmonary macrophages and B cells [20], we utilized WLLC suspension samples in our study. Our findings illuminate distinct PR responses and differential gene expressions related to metabolic and circadian rhythm targets during MDV infection, particularly when comparing chicken breeds and shedder hosts.

## 2. Materials and Methods

### 2.1. Ethics Statement

All animal procedures were preapproved by the UIUC’s Institutional Animal Care and Use Committee (IACUC) and were conducted according to national regulations and ARRIVE guidelines. The animal care facilities and programs of UIUC meet all the requirements of the law (89–544, 91–579, 94–276) and NIH regulations on laboratory animals and comply with the Animal Welfare Act, PL 279, and are accredited by the Association for Assessment and Accreditation of Laboratory Animal Care (AAALAC). All experimental procedures complied with approved IACUC protocols. Water and food were provided *ad libitum*.

### 2.2. Experimental Approach

The current study represents a continuation of our recently published work [9]. Therefore, similar experimental approaches were adopted with minor modifications per the current study’s requirements. Briefly, day-old chickens from two different chicken lines, MD-susceptible Pure Columbian (PC, n = 40) and MD-resistant White Leghorns (WL, n = 40) were acquired from the UIUC Poultry Research Farm and initially housed separately. Three-day-old chicks (n = 12 per chicken line) were experimentally infected by intraabdominal inoculation of 2000 plaque-forming units (PFU) with the dual fluorescent v2001 (RB-1B strain) previously described [9]. After 2 weeks, MDV was shed from experimentally infected chickens termed “shedders.” Infection of shedders was confirmed based on visualizing feathers for fluorescently tagged MDV (pUL47eGFP and RLORF4mRFP expression) as described previously [21,22]. 

It has been shown that sentinel contact birds can be efficiently infected when exposed to shedders for as little as 4 h [19]. We performed an “exposure” experiment where age-matched chickens were added to the experimentally infected shedders to address the early or short-acting PR response during natural MDV infection. Four mixed (PC, n = 4; WL, n = 4) sets of naïve age-matched chickens were separately exposed to PC (×PC) and WL (×WL) shedders for either 6 or 24 h post-infection (pi), at which point the birds were euthanized. Another two sets (one set per chicken line) of age-matched uninfected chickens (n = 4) were held in two separate rooms to be used as uninfected control groups (Figure 1).

### 2.3. Sample Collection and RNA Extraction

To obtain single-cell suspensions of lymphocytes and mononuclear cells from the lungs (WLLC), our published protocol was used [9]. Briefly, chickens were humanely euthanized and immediately perfused with PBS via oral gavage, and WLLC suspensions were pelleted via centrifugation at 1000× g for 10 min at 4 °C and then frozen at −80 °C. These pellets were resuspended in 500 µL RNA-STAT60 and processed for total RNA extraction using the manufacturer’s instructions (Tel-Test, Inc., Friendswood, TX, USA). RNA concentrations were measured using a Nano-Drop ND-1000 spectrophotometer (Nano-Drop Technologies, Wilmington, DE, USA). The purity of RNA (A260/A280) for all samples was above 1.81, and the quality was evaluated using 2% agarose gel. High-quality samples were used for cDNA synthesis. 

### 2.4. Primers

The procedure for primer design, primer testing, selection of internal control genes for normalization, and detail for qPCR analysis were as previously reported [23,24]. Briefly, gene specific primers were designed using gene bank mRNA sequences for P1 PRs [P1A1A (ADORA1), P1A2A (ADORA2A), P1A2B (ADORA2B), P1A3 (ADORA3)], P2X PRs (PX1- P2X7) and P2Y PRs (P2Y1, P2Y2, P2Y3/P2Y6, P2Y4, P2Y5, P2Y8, P2Y10, P2y12, P2Y13, P2Y14), and other target cellular genes DDX5, BCL2, ANGPTL4, STAT1, STAT5A, CAT, SOD1 GPX1, and GPX2. Detailed information about the primers is provided in Table 1.

### 2.5. RT-qPCR Analysis

RT was performed using 2 µg Turbo DNA-free (Thermo Fisher Scientific, Waltham, MA, USA)-treated total RNA using the High-Capacity cDNA Reverse Transcription Kit (Thermo Fisher Scientific). Twenty-microliter RT reactions were carried out according to the manufacturer’s instructions with Oligo dT20 plus random primers. The reaction mixture was incubated at 25 °C for 10 min, then 37 °C for 120 min, followed by 85 °C for 5 min. Most cDNA reactions were diluted 1:4 in ddH2O before qPCR analysis.

For qPCR, 20 µL reactions were prepared with 4 µL diluted cDNA, specific primers (0.5 μM), 10 µL 2× Power SYBR Green Master Mix (Thermo Fisher Scientific), and ddH2O as previously published [23,24]. Serial 10-fold dilutions of pooled cDNA of the respective samples generated standard curves, starting with approximately 500 pg of cDNA. Thermal cycling conditions were as follows: 50 °C for 2 min and 95 °C for 10 min, followed by 40 cycles at 95 °C for 15 s and 60 °C for 1 min. All RT-qPCR assays were performed using an Applied Biosystems QuantStudio 3 real-time PCR system (Thermo Fisher Scientific), and the results were analyzed using QuantStudio Design & Analysis Software v1.4.2, supplied by the manufacturer. The final gene data set was normalized with a geometric mean of chicken GAPDH. 

### 2.6. Statistical Analysis

Statistical analyses were performed using IBM SPSS Statistics version 28 software (SPSS Inc., Chicago, IL, USA). The normalized RT-qPCR gene expression data were analyzed using three-way ANOVA followed by Tukey’s post hoc tests: virus (V), chicken breed (B), time (T), and all possible interactions (V × B), (V × T), and (B × T) were used as fixed effects, and the relative mRNA were used as dependent variables. Statistical significance was declared at *p* < 0.05, and the mean tests associated with significant interactions (*p* < 0.05) were separated using Tukey’s tests. The comparisons were made between the groups, including age-matched control and exposed groups. Superscripts (a–e) were used to indicate significant differences between the groups.

## 3. Results and Discussion

During natural MDV infection, the recruitment and infection of various immune cells, including macrophages and B and T lymphocytes, play a crucial role in MDV replication and disease progression [25]. Unlike mammals, avian macrophages are not abundant on the outer surface of the lung’s airway epithelia [26,27,28]. However, upon exposure to infections, these macrophages and other immune cells are drawn to the surface, triggering a cellular immune response within the avian respiratory tract [28,29]. Therefore, our study focused on sampling the lung airway during natural infection and utilized WLLC, an efficient method for collecting superficial, loosely attached cells from the lung epithelia [9]. 

To assess the relative expressions of three subtypes of PRs, we studied two chicken lines exposed to MDV-infected PC (MD-susceptible) and WL (MD-resistant) chickens for 6 and 24 h, comparing mRNA expressions with uninfected controls from the same breed. Based on breed, time, and infection source, we categorized the exposed birds into eight groups: WL and PC chickens exposed to PC-infected birds for 6 and 24 h (×PC) and WL and PC exposed to WL-infected environments for 6 and 24 h (×WL)—uninfected birds for both chicken lines served as uninfected, unexposed controls.

### 3.1. Confirmation of MDV Infection

To confirm that birds exposed to shedders for 6 or 24 h were infected, we used RT-pPCR assays to detect viral gene expression in WLLC. Viral mRNA was not detected in the lungs of uninfected controls, while expression of the late MDV gene, UL44 encoding glycoprotein C (gC), was detected at 6 h and increased at 24 h (Figure 2), indicating initiation of infection in those birds. This is consistent with previous findings of MDV establishing infection by 4 h [19].

### 3.2. Non-Purinergic Receptor Regulatory Gene Responses

#### 3.2.1. DDX5

DDX5 is a helicase and has been reported as a critical element for recognizing viral nucleic acid, regulating viral replication and infection-related signaling, and modulating antiviral innate immunity and is therefore considered a potential antiviral candidate for many viral infections [30]. We observed a significant increase in DDX5 expression at 6 h in MDV-infected WL and PC chickens exposed to PC birds compared with age-matched controls. The increase was even more pronounced in PC chickens at 24 h, while it was back to basal levels for WL chickens (Figure 3a). The innate immune system acts as the body’s first line of defense against viral infections by identifying different patterns associated with pathogens [31]. Upon viral invasion, specific receptors in the immune system recognize these patterns, initiating signaling pathways that activate proteins like TANK-binding kinase 1 (TBK1) and interferon regulatory factor 3 (IRF3) [32,33]. This activation leads to the production of proinflammatory cytokines and type I interferons (IFN-I). These interferons, particularly IFN-α and IFN-β, then prompt the expression of IFN-stimulated genes (ISGs) and chemokine (C-X-C motif) ligand 10 (CXCL10), among others. These molecules play vital roles in thwarting viral spread and eliminating infected cells, ultimately helping to control the viral infection [34,35,36]. Zan and colleagues [37] demonstrated that DDX5 modulates the body’s innate immune response against viral infections by interacting with a specific protein called serine/threonine-protein phosphatase 2 A catalytic subunit beta (PP2A-Cβ), leading to the deactivation of IRF3. When DDX5 levels are reduced, there is a notable increase in the production of IFN-I and the expression of ISGs in response to DNA or RNA virus infections. Conversely, when DDX5 is artificially introduced, it suppresses IFN-I production and enhances viral replication [37]. In the current context, the increase in *DDX5* may be associated with protecting the virus against the innate immune response; however, further research is required to address this result’s importance.

#### 3.2.2. BCL2

BCL2 is an antiapoptotic member of the Bcl-2 protein family [38]. It is not only vital in cells, but numerous viruses have hijacked this gene, including adenoviruses, poxviruses, Epstein–Barr virus (EBV), γ-herpesviruses 68, Kaposi sarcoma-associated herpesvirus, and turkey herpesvirus [39] that mimic host Bcl-2 proteins and hijack the intrinsic apoptotic pathway for their benefit [40,41]. In the current study, the naturally infected chickens from both breeds showed higher expression of BCL2 irrespective of the shedder breed (Figure 3b). This may be related to a recent report showing that MDV infection extends the survival of B cells in culture and protects B cells from apoptosis [42], suggesting this is a critical MDV-induced response during natural infection. Another study also reported a decrease in neural death with an expression of BCL2 from defective HSV-1 vectors [43], whereas a negative correlation has been reported between MDV latency and apoptosis in the spleen [44]. Although the infection in the lungs is poorly understood, it is presumed that following initiation of infection, antigen-presenting cells are recruited to the lungs, including B cells [45], and that increased expression of BCL2 may increase B cell survival [42].

#### 3.2.3. ANGPTL4

Angiopoietin-like 4 (ANGPTL4) is widely known for its role as an adipokine in regulating metabolism. However, recent advances in understanding ANGPTL4 in diseases have revealed novel insights into other aspects, such as cancer progression and therapeutics [46]. Moreover, this gene has been reported to increase lung leakiness, exacerbating inflammation-induced lung damage [47]. Our current study observed an increase in *ANGPTL4* in all the naturally infected groups (Figure 3c).

#### 3.2.4. STAT1A and STAT5A

The expression of *STAT1A* increased, especially in PC chickens at 6 and 24 h, whereas relatively minor changes were observed in WL chickens (Figure 3d). Compared to the age-matched control groups, all infected birds showed significantly higher expression of *STAT5A* at 6 and 24 h (Figure 3e). Overall, PC chickens exhibited a pronounced expression at 6 h post exposure. STAT1A is among the major mediators of the cellular immune response to IFNs and is a key regulator of the antiviral immune response [48]. Both pro- and anti-apoptotic responses of IFN receptors are initiated via STAT1 or STAT3 and STAT5AB, respectively, where the STAT1-dependent cascade accounts for apoptosis while STAT3 and STAT5 promote survival [49]. The increase in the expression of these genes can also have apoptotic and survival effects. Furthermore, these genes have been reported to promote adaptive natural killer responses to herpesvirus infections [50,51]. The elevated levels of *STAT5A* transcripts can also be associated with increasing the survival and homeostasis of antigen-presenting cells [51,52,53], which is one of the main requirements for MDV replication. The increase in *BCL2* expression also supports the observed increase because *STAT5A* has been reported to enhance survival, AKT activation, and *BCL2* expression [54].

#### 3.2.5. Oxidative Stress Markers

The glutathione peroxidase family (GPX) works with superoxide dismutase (SOD) and catalase (CAT) to form the enzymatic antioxidant cascades against oxidative stress, thereby limiting its toxicity [55,56]. The expression of oxidative stress markers (Figure 3f–i) significantly increased in MDV-infected birds at 6 h but then decreased at 24 h, although it remained significantly higher than in the control group. A consistent increase was observed in WL birds, except those exposed to MDV-infected PC chickens (×PC) for 6 h. It is well established that viral infection induces the production of reactive oxygen molecules in the host, and a strong connection has been reported between viral infection and oxidative stress development [57,58,59]. Moreover, viral infection suppresses the antioxidative defense mechanisms of the host, influencing pathogenesis, disease progression, and severity [59,60,61]. MDV has been shown to alter the oxidative status of birds similar to other poultry viruses, including duck hepatitis virus B, NDV, and avian influenza virus [62,63], while other studies observed a decrease in the activities of oxidative stress markers [62,63,64]. The increase in the relative mRNA expression of these markers (Figure 3f–i) indicates an early host response to infection, as observed in other study models [65], whereas the previously reported decrease in these markers’ activities is associated with the later stage of disease progression.

#### 3.2.6. Summary of Target Gene Responses

Oxidative stress has been linked to MD [66,67], affecting various organs like the spleen, thymus, and liver by altering the expression of SOD, CAT, and GPX [62]. Increased DDX5 expression (Figure 3a) suggests a role in viral protection from the host immune system, while higher BCL2 levels (Figure 3b) indicate B cell recruitment, which is important for MDV infection. Elevated STAT5A (Figure 3e) supports antigen-presenting cell survival, and increased ANGPTL4 (Figure 3c) may facilitate MDV replication and transmission by altering metabolic functions and causing lung leakage. The rise in oxidative stress markers SOD1, CAT, GPX1, and GPX2 (Figure 3f–i) signals an early host response to infection, as observed in other study models [65], in contrast to the reported decrease in later stages of the disease. 

### 3.3. P1 PR Responses during Natural Infection

#### 3.3.1. P1A1

P1A1 (Adenosine A1 Receptor: ADORA1) is widely expressed in tissues, and its role has been reported to be anti-inflammatory, anti-diuretic, and involved in tissue protection, particularly the lungs and kidneys [68]. It has also been studied during EBV infection. Du et al. [69] showed that cordycepin (3-deoxyadenosine), an adenosine derivative with anti-proliferative, anti-inflammatory, and pro-apoptotic effects, induced EBV reactivation in EBV-transformed cells. However, Ryu et al. [70] showed that cordycepin suppressed EBV replication. In WLLC, *P1A1* expression increased in WL chickens (*p* < 0.05) exposed for 6 and 24 h to experimentally infected WL (×WL) compared to control and other groups, while a relatively lower expression was measured in the 6 and 24 h infected PC groups (Figure 4a).

In our previous study, we also reported increased *ADORA1* (*P1A1*) in the WLLC samples from diseased and infected WL chickens > 4 wk pi [9]. *ADORA1* expression in WLLC in our previous and current study (Figure 4a) suggests that ADORA1 could be associated with tissue protection. Choi and colleagues [71] demonstrated that adenosine partially requires ADORA1 signaling to upregulate BZLF1, a key regulator of EBV lytic replication. Furthermore, they also confirmed that BZLF1 upregulation by adenosine effectively suppresses or delays EBV-associated gastric carcinoma development [71]. With the data reported here, the higher expression of *ADORA1* in the WLLC of infected WL chickens (Figure 4a) suggests they may have a more robust anti-inflammatory reaction that could be involved in their being more resistant to MD development. Further experiments are warranted to understand better the role ADORA1 plays during MDV infection and viral pathogenesis. 

#### 3.3.2. P1A2A and P1A2B

No change was observed in the expression of *P1A2A* in both PC and WL naturally infected groups (Figure 4b), whereas *P1A2B* showed higher expression in PC or WL chickens infected from PC chickens (×PC) at 6 h (Figure 4c). P1A2B (Adenosine A2B Receptor: ADORA2B) and P2Y2 have been reported to regulate mucociliary clearance, a dominant component of pulmonary host defenses [72]. Higher expression of these receptors in the WLLC of naturally infected WL chickens suggests it may play a unique role in the lungs of these chickens. Interestingly, P1A2B signaling has been reported to reduce oxidative bursts from immune cells [50,73,74]. Our current study observed an increased expression of oxidative stress markers (Figure 3: *SOD1*, *CAT*) in naturally infected PC chickens. These results suggest a potential role of P1A2B in reducing oxidative stress in WL chickens during initial infection compared to PC chickens. That is, P1A2B signaling as anti-inflammatory and protection during stress and disease conditions reported by different studies [75] could play a role during MD progression in susceptible PC chickens. 

#### 3.3.3. P1A3

P1A3 (Adenosine A3 Receptor: ADORA3) has been demonstrated to mediate anti-inflammatory, anti-cancer, and anti-ischemic protective effects. P1A3 is overexpressed in cancer and inflammatory cells, while lower expression is found in normal cells [76]. In cancer cells, the activation of the P1A3A corrects an imbalance in the downstream Wnt signaling pathway [77,78]. Administration of a P1A3 agonist to activate its cell surface receptor inhibits the formation of cAMP and indirectly decreases phosphorylation and therefore decreases inactivation of the serine/threonine kinase GSK-3β. The resulting increased phosphorylation of β-catenin results in it being removed from the cytoplasm by ubiquitination, thereby preventing its nuclear import and inhibiting cell growth. Concerning cancer, nuclear factor κB (NF-κB) is a potent anti-apoptotic agent in malignant cells, and its activation is strongly associated with tumors [79,80]. Whereas regulation of NF-κB has been reported as central to MDV-induced pathogenesis [81], P1A3A induces specific anti-inflammatory and anti-cancer effects via a molecular mechanism that entails modulation of the Wnt and the NF-κB signal transduction pathways [77,82,83]. Interestingly, *P1A3* in WL chickens exposed for 6 and 24 h to PC-infected chickens (×PC) highlights the importance of the receptors as immediate early PRs (Figure 4d).

#### 3.3.4. Summary of P1 PR Responses

MDV infection induces inflammatory cytokines in the lung’s epithelium [84] and P1A2B and P1A3 functions are associated with anti-inflammatory responses in the lung [85,86]. The higher mRNA expression of these receptors in WLLC samples (Figure 4c,d) at 6 h in WL chickens suggests an anti-inflammatory response to higher stress. The higher expression of P1A2B and P1A3 in these birds’ lungs may help regulate the active transport across the epithelium during disease stress. Overall, the differential expression of P1 PRs (P1A1, P1A2B, and P1A3) at early points of natural infection suggests these receptors could also be important to overcome oxidative stress and produce anti-inflammatory conditions. Interestingly, *P1A2B* and *P1A3* expression in WL chickens at 6 and 24 h pi highlight the importance of these receptors as immediate early PRs, where both have been reported as part of adenosine-mediated anti-inflammatory pathways [76,86].

### 3.4. P2X PR Responses during Natural Infection

#### 3.4.1. P2X1, P2X2, and P2X3

Evidence suggests that P2X PRs play crucial roles in host defense against infections, indicating their importance in these cells [87]. PC chickens exhibited higher expression of *P2X1*, whereas WL chickens had a lower expression when exposed to WL-infected chickens (×WL) infected for 24 h (Figure 5a). Lee et al. [6] observed increased P2X PR expression in EBV-infected B cells. These receptors can form homo- and heteromeric (e.g., P2X2/P2X3) ion channels for different purposes, depending on the cell type they are expressed in and the intended response [88]. Furthermore, P2X receptors containing P2X2 and P2X3 subunits are crucial in responding to hypoxia [89]. The upper and lower respiratory tracts exhibit different P2X3 distributions, indicating varied responses [90,91,92]. While *P2X3* expression was not detected in the liver in our prior study, a significant increase was noted in WLLC in both chicken lines late during infection (>4 wk pi) [9]. The heightened expression of *P2X3* and *P2X2* in the lungs (Figure 5b,c) suggests that MDV infection triggers this response, which is expected to be more pronounced in PC birds than in WL.

#### 3.4.2. P2X5

Our study revealed increased *P2X5* expression in both MDV-infected WL and PC WLLC samples (Figure 5e), significantly more so in WL chickens, aligning with our previous observations later during natural infection (<4 wk pi) [9]. This underscores the importance of the P2X5 response in early and late infection stages. The observations from both studies support Chen et al. [12] results where inhibition of P2X5 inhibited HCMV replication, and Lee et al. [6] observed higher P2X5 expression in EBV-transformed cells. P2X5 is vital in mounting proper innate immune responses by regulating ATP-mediated inflammasome activation and IL-1β production during infection [93,94]. Comparing data from our previous and current studies suggests it may be associated with immune regulation of virus replication.

#### 3.4.3. P2X7

The decrease in *P2X7* in WL chickens (Figure 5g) was interesting since we observed significant increases in *P2X7* in disease groups in our previous study [9], which suggested greater importance in the later stages of MD progression. This hypothesis is supported by data liking P2X7 expression to disease progression due to its role in regulating alveolar macrophage activation, secretion of IL-1β and IL-1α, and neutrophil recruitment, inflammasome, caspases, and phospholipases [95,96,97,98]. In addition, P2X7 has also been reported to modulate intracellular signaling pathways, such as PI3K/AKT/mTOR, myeloid differentiation factor 88 (MyD88)/NF-κB, and the activation of mitogen-activated protein kinase (MAPK) pathway proteins (MEK, ERK 1/2) [99,100,101]. Our data here suggest its downregulation may be important during the initiation of infection and then later during disease progression.

#### 3.4.4. P2X4 and P2X6

We found no P2X4 and P2X6 expression changes in both chicken lines (Figure 5d,f).

#### 3.4.5. Summary of P2X PR Responses

P2X receptors are ion gate channels that quickly respond to stimuli and play a critical role in host defense against infections [87,102]. In the current study, the noticeable increase in *P2X2* and *P2X3* receptor expression highlights the importance of these receptors during early lung infection, especially in PC birds. *P2X4* (Figure 5d) expression remained unchanged, indicating its potential role in MDV pathogenesis as reported earlier [9]. The relative increase in *P2X5* (Figure 5e) expression in both WL and PC birds, with a more significant increase in WL, suggests its importance in both early and late infection stages, supported by our previous observations [9]. Meanwhile, the decrease in *P2X7* (Figure 5g) expression in WL birds suggests its role in the later stages of MD progression, as we observed significant changes in disease groups in our previous study [9].

### 3.5. P2Y PR Responses during Natural Infection

#### 3.5.1. P2Y1

P2Y1 is widely distributed in tissues [103] and has been reported to play roles in innate and adaptive immune responses by inducing endothelial cell activation and leukocyte rolling [104,105]. Our earlier study [9] showed lower expression of *P2Y1* in WLLC in both chicken lines and higher hepatic expression in PC chickens > 4 wk pi. In contrast, WL chickens infected for 6 h had significantly higher expression of *P2Y1* (Figure 6a) when exposed to MDV-infected WL (×WL) and PC (×PC) birds. However, the considerably higher expression when housed with PC birds suggests the viral load of exposure may be important as PC chickens are more susceptible and shed higher virus levels. Endothelial cells, crucial components of the lungs [106], have also been reported to express P2Y1 [107,108,109]. The current observation implicates P2Y1 in the lungs, potentially ameliorating protein leakage and enhancing anti-inflammatory responses [110]. This hypothesis is supported by the increased expression of *ANGPTL* (Figure 3c), a gene that increases lung leakiness, exacerbating inflammation-induced lung damage [47]. Further research is needed to address the role of P2Y1 in natural MDV infection in the lungs.

#### 3.5.2. P2Y3/P2Y6

The expression of *P2Y3* (avian homolog of mammalian P2Y6) [111] was differentially expressed in infected PC chickens at 6 h, depending on the MDV-infected chicken line the birds were exposed to (Figure 6c). That is, *P2Y3* expression was decreased when WL chickens were housed with MDV-infected PC chickens (×PC) but increased when WL birds were housed with MDV-infected WL chickens (×WL). PC chickens had increased *P2Y3* expression when housed with MDV-infected PC (×PC) and WL (×WL) birds at 24 h exposure. For mammals, P2Y6 has been reported to contribute to airway inflammation following mice’s allergic response induction [112]. The antiviral role of P2Y6 has been demonstrated in different studies under different viral infections. For example, inactivated avian influenza virus-H5N1 increases *IL-6* and *CXCL8* mRNA by a mechanism that involves the activation of P2Y6 [113]. In addition, vesicular stomatitis virus (VSV)-induced cell death and virus replication were enhanced significantly by knocking down or out P2Y6 in different cells [114]. The current data and our previous data on P2Y3/P2Y6 in viral replication are further supported by high expression of *BCL2* (Figure 3b), which would be expected to increase the survival of the infected cell and, ultimately, viral load. These results warrant further research to understand this receptor’s potential role during infection and alteration of viral load.

#### 3.5.3. P2Y5

The present study showed a significant increase in *P2Y5* (20 ± 5 fold-change) in WL chickens only (Figure 6d). In our earlier study, the same breed exhibited higher expression in WLLC at >4 wk pi, suggesting the importance of P2Y5 in protecting WL chickens from the pathological effects of MDV. This G protein-coupled receptor (GPCR) has been reported to bind and be activated by lysophosphatidic acid (LPA) and farnesyl pyrophosphate (FPP) [115]. Elevated LPA levels are typically observed during pathological conditions [116]. It has been noted that higher levels of LPA can regulate the Gα12/13-Rho/ROCK pathway via LPAR4/P2Y9 and LPAR6/P2Y5 [117]. The Rho-ROCK pathway regulates MDV cell-to-cell spread in cell culture [118]. Moreover, LPA maintains epithelial integrity by regulating cellular processes such as the renewal and migration of epithelial cells and inflammation responses [119,120]. LPA-mediated induction has also been shown to regulate the activity of p38 MAPK, PI3K, PLC, and PKC and induce ERK1/2 phosphorylation [115].

#### 3.5.4. P2Y8 

Similar to our previous study, which showed differential expression of *P2Y8* in PC-infected birds later during infection, our present study showed a significant increase in PC birds early during infection (Figure 6e). P2Y8 exhibits higher lymphocyte expression, whereas lower expression is observed in the lungs and other visceral organs [121]. This receptor has been implicated in oncogenesis and can form fusions with other proteins, such as cytokine receptor-like factor 2 (CRLF2), in various diseases [122,123,124]. Based on our current observation and the potential role of P2Y8 in oncogenesis, it is compelling to support our previous speculation [9] that P2Y8 may play an important role in MDV pathogenesis in the PC chicken line.

#### 3.5.5. P2Y10

The increased expression of *P2Y10* (Figure 6f) is noteworthy due to its presence in immune cells such as B and T cells, monocytes, dendritic cells, and granulocytes [125,126]. P2Y10 is regulated by nucleotides, LPA, sphingosine-1-phosphate (S1P), and lysophosphatidylserine (LysoPS) [127,128,129,130], and it facilitates the activation of RhoA in CD4+ T cells, thus mediating chemokine-induced migration and contributing to T cell-mediated diseases [130]. The increase observed in the current study (Figure 6f), as well as our earlier report [9], supports the importance of P2Y10 in disease progression and MD susceptibility.

#### 3.5.6. P2Y13

Among the PR family, P2Y13 stands out as one of the most critical PRs expressed in the lungs [131] and restricts the replication of both DNA (HSV-1) and RNA (NDV and VSV) viruses via JAK-STAT signaling in vitro [15]. Additionally, the receptor has been reported to improve recurrence-free survival in hepatocellular carcinoma patients [132]. In the current study, the higher expression of *P2Y13* during earlier infection (Figure 6g), combined with the data from our previous study [9], underscores the potential significance of P2Y13 in MDV pathogenesis.

#### 3.5.7. P2Y14

Like our previous study [9], we observed an increase in *P2Y14* expression in the WLLC from PC birds regardless of the source of infection (Figure 6h). This finding supports the potential role of the P2Y14 receptor in MDV replication and pathogenesis, as reported in other diseases [133]. Uridine diphosphate (UDP)-glucose (UDPG) has been shown to promote the recruitment of neutrophils and macrophages in the lung [134]. Additionally, UDPG and UDP-glucose sugars (UDPG-sugars) act as potent regulators of P2Y14, initiating subsequent signal transduction pathways via Gi/o-coupled proteins [135]. The observed increase in *ANGPTL4*, which increases lung leakiness [47], and *BCL2*, which extends the survival of infected B cells [40,41], coupled with the regulation of BCL2 expression by P2Y14 [136], suggests that MDV increases the number of macrophages in the lungs of infected chickens with the help of ANGPTL4 [84]. Moreover, it increases the survival of B cells by protecting them from apoptosis through the regulation of BCL2 via P2Y14 [42]. Furthermore, *P2Y14* is increased by MDV infection, leading to increased IFN-γ stimulation, which promotes glycogen synthesis in macrophages [137]. This process is channeled through glycogenolysis to generate glucose-6-phosphate (G6P) and NADPH, ensuring high levels of reduced glutathione for the survival of inflammatory macrophages [138]. Consequently, this increases UDPG levels and P2Y14 expression in macrophages. These findings collectively suggest a crucial role for P2Y14 in protecting MDV-infected B cells and potentially in recruiting macrophages or other immune cells to the lungs.

#### 3.5.8. Summary of P2Y PR Responses

P2Y receptors regulate immune cell functions, including phagocytosis and pathogen killing [139]. Some P2Y receptors are potential therapeutic candidates against viral infections, including herpes viruses [15,114]. In the current study, *P2Y1* (Figure 6a) showed significantly higher expression at 6 h in WL birds, indicating its role in immunoregulation and response varies with viral load. *P2Y2* (Figure 6b) showed no change, reflecting tissue specificity supported by the higher hepatic expression in our previous study [9]. *P2Y3* (Figure 6c) responses suggest its importance in viral infection and replication by increasing infected cell survival. Higher *P2Y5* (Figure 6d) expression in WL birds suggests a protective role against MDV, whereas significant increases in *P2Y8*, *P2Y10*, *P2Y13*, and *P2Y14* (Figure 6e–h) expression in PC birds, supported by previous data [9], highlight their importance in MDV pathogenesis.

## 4. Conclusions

In our current study, we measured the PR response during the initiation of MDV infection in the lungs. Our findings suggest a significant role of PRs in genetic resistance to the progression of MD where we observed differential expression for many PRs between MD-resistant and -susceptible chickens during natural infection. In the MDV-resistant breed (WL), we noted significant changes in the expression of P1 PRs involved in tissue protection upon infection initiation. Additionally, higher levels were observed for *P2X5*, *P2Y1*, and *P2Y5* expression. Conversely, in MD-susceptible PC chickens, the changes in expression of *P2X2*, *P2Y8*, *P2Y10*, *P2Y13*, and *P2Y14* indicate their role in reduced protection to induction of MD in these chickens. 

As expected, there were differences in virus replication in the two chicken lines early in the infection. PC chickens had significantly higher levels of MDV replication in the lungs, regardless of the shedder chicken line (Figure 2). Thus, it is difficult to determine whether the differential expression of host genes is directly responsible for the differences in MDV replication between PC and WL chickens or indirectly related to the different levels of virus infection. However, the differential expression suggests they play a role in MDV infection in the lungs and likely the progression of the disease. These current results are compelling and suggest the need for further studies to elucidate the genetic differences in PR responses to MDV infection.

## Figures and Tables

**Figure 1 viruses-16-01130-f001:**
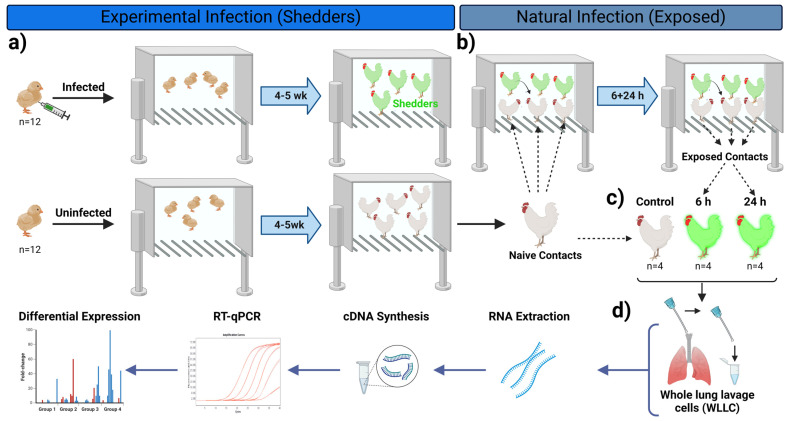
Visual presentation of our experimental model system to examine early gene expression during lung MDV infection. PC and WL chickens were experimentally infected with cell-associated MDV and housed separately. After 2 weeks, experimentally infected chickens began to shed the virus and were termed “shedders”. Two groups of naïve mixed chicken breeds (n = 8/breed) were exposed to the shedders. One group to PC Shedders (×PC) and the other to WL Shedders (×WL) to initiate infection through the natural route. WLLC samples were collected and used in RT-qPCR assays. Figure generated with Biorender.com (accessed on 10 June 2024).

**Figure 2 viruses-16-01130-f002:**
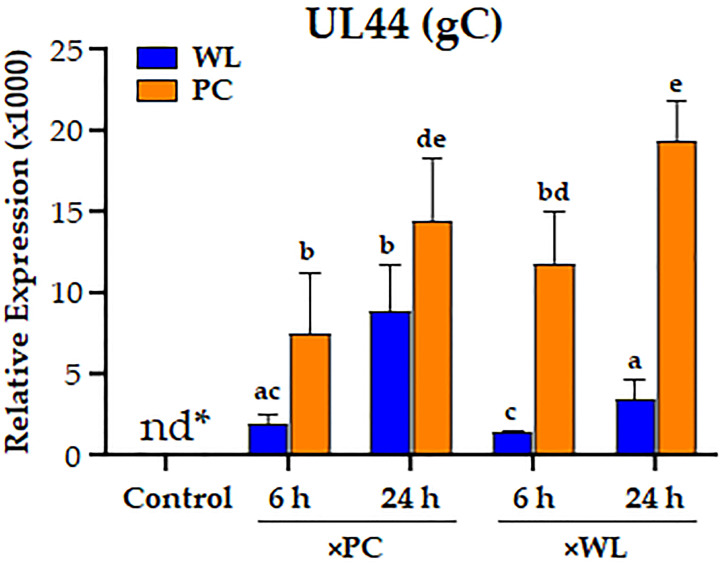
The relative mRNA expression of MDV UL44 (gC) in the MD-resistant (WL) and MD-susceptible (PC) chicken lines exposed to MDV-infected shedders (×PC and ×WL) compared with uninfected control (nd* = not detected). Data are presented as mean ± SD. Statistical significance was calculated using Multivariate ANOVA with factors chicken line (B), infection (I), and time (T), followed by Tukey’s post hoc test. Groups with different superscripts (a–e) are significantly different (*p* < 0.05) from each other. * = not detected.

**Figure 3 viruses-16-01130-f003:**
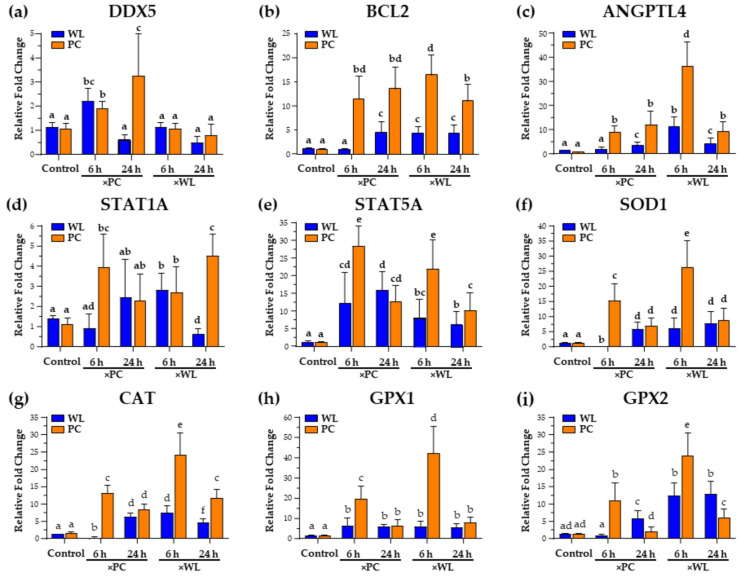
The relative mRNA expression of *DDX5* (**a**), *BCL2* (**b**), *ANGPLT4* (**c**), *STAT1A* (**d**), *STAT5A* (**e**), *SOD1* (**f**), *CAT* (**g**), *GPX1* (**h**), and *GPX2* (**i**) in the MD-resistant (WL) and MD-susceptible (PC) chicken lines exposed to MDV-infected shedders (×PC and ×WL) was compared to uninfected controls. Data are presented as mean ± SD. Statistical significance was calculated using Multivariate ANOVA with factors chicken line (B), infection (I), and time (T), followed by Tukey’s post hoc test. For specific genes, groups with different superscripts (a–e) are significantly different (*p* < 0.05) to each other.

**Figure 4 viruses-16-01130-f004:**
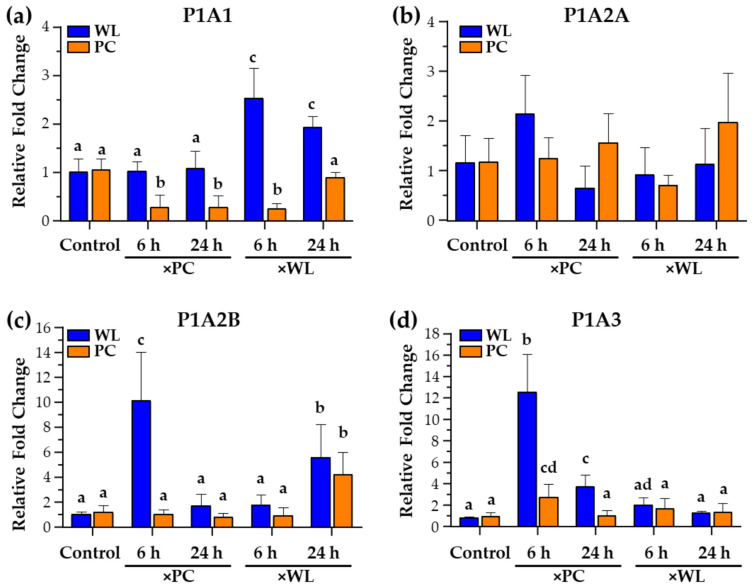
The relative mRNA expression of chicken P1 PR genes *P1A1* (**a**), *P1A2A* (**b**), *P1A2B* (**c**), and *P1A3* (**d**) in the MD-resistant (WL) and MD-susceptible (PC) chicken lines exposed to MDV-infected shedders (×PC and ×WL) was compared to uninfected controls. Data are presented as mean ± SD. Statistical significance was calculated using Multivariate ANOVA with factors chicken line (B), infection (I), and time (T), followed by Tukey’s post hoc test. For specific genes, groups with different superscripts (a–d) are significantly different (*p* < 0.05) to each other.

**Figure 5 viruses-16-01130-f005:**
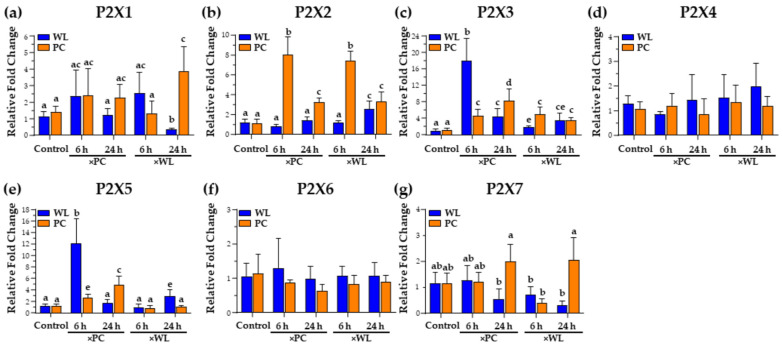
The relative mRNA expression of chicken P2X PR genes *P2X1* (**a**), *P2X2* (**b**), *P2X3* (**c**), *P2X4* (**d**), *P2X5* (**e**), *P2X6* (**f**), and *P2X7* (**g**) in the MD-resistant (WL) and MD-susceptible (PC) chicken lines exposed to MDV-infected shedders (×PC and ×WL) was compared to uninfected controls. Data are presented as mean ± SD. Statistical significance was calculated using Multivariate ANOVA with factors chicken line (B), infection (I), and time (T), followed by Tukey’s post hoc test. For specific genes, groups with different superscripts (a–e) are significantly different (*p* < 0.05) to each other.

**Figure 6 viruses-16-01130-f006:**
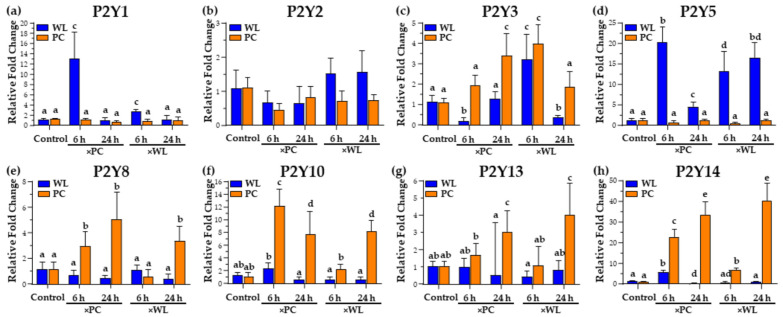
The relative mRNA expression of chicken P2Y PR genes *P2Y1* (**a**), *P2Y2* (**b**), *P2Y3* (**c**), *P2Y5* (**d**), *P2Y8* (**e**), *P2Y10* (**f**), *P2Y13* (**g**), and *P2Y14* (**h**) in the MD-resistant (WL) and MD-susceptible (PC) chicken lines exposed to MDV-infected shedders (×PC and ×WL) was compared to uninfected controls. Data are presented as mean ± SD. Statistical significance was calculated using Multivariate ANOVA with factors chicken line (B), infection (I), and time (T), followed by Tukey’s post hoc test. For specific genes, groups with different superscripts (a–e) are significantly different (*p* < 0.05) to each other.

**Table 1 viruses-16-01130-t001:** Primers used for RT-qPCR assays.

Gene Name	Gene ID ^1^	Accession No. ^2^	Primer ^3^	Sequence (5′–3′)	Fragment Size ^4^
*DDX5*	395629	NM_204827.2	F. 585	AACTCGTGAACTGGCCCAAC	228
			R.812	TCCGCTTCATCAAGGACGAG	
*BCL2*	396282	NM_205339.3	F. 3227	CACAGGTGCCTACTGTCGTT	225
			R. 3451	CACACTGGGATTCTTCCGCT	
*ANGPTL4*	769087	XM_040692473.2	F. 1134	AGCCTACACACTCAACCTGC	388
			R. 1521	ACCAACTGATGGAGCTCACG	
*STAT1A*	424044	NM_001012914.2	F. 2121	CCCAAGGGAAACGGCTACAT	461
			R. 2581	CACTGAGGACCCCTTGTTCC	
*STAT5A*	395556	NM_204779.1	F. 1303	CAGACCAAGTTTGCAGCCAC	356
			R. 1658	GACAGCGTCTTCACCTGGAA	
*SOD1*	395938	NM_205064.2	F. 117	TCATCCACTTCCAGCAGCAG	333
			R. 449	CCCCTCTACCCAGGTCATCA	
*CAT*	423600	NM_001031215.2	F. 1338	GCGCCCCGAACTATTATCCA	280
			R. 1617	ATACGTGCGCCATAGTCAGG	
*GPX1*	100857115	NM_001277853.3	F. 422	GATGACCAACCCGCAGTACA	315
			R. 736	AGCTTTGAAAACATCGGGCG	
*GPX2*	100857454	NM_001277854.3	F. 31	CGCCAAGTCCTTCTACGACC	133
			R. 163	GGTGTAATCCCTCACCGTGG	

^1^ National Center for Biotechnology Information (NCBI) Gene ID for *Gallus gallus* (chicken). ^2^ NCBI mRNA Accession No. ^3^ Primer direction (F-forward; R-reverse) and hybridization position in gene. ^4^ Amplicon size in base pairs.

## Data Availability

All data are contained within the article.

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
