# Peer review of "Temporal Dynamics of Purinergic Receptor Expression in the Lungs of Marek’s Disease (MD) Virus-Infected Chickens Resistant or Susceptible to MD"

_viruses, 2024, doi:10.3390/v16071130_

Round 1

Reviewer 1 Report

Comments and Suggestions for Authors

In the present study, the authors analyzed the expression patterns of PR genes in MD-resistant and -susceptible chickens in lungs at early infection. In addition, the authors compared the expression levels of candidate molecules involved in the pathogenesis between the groups. The altered expression was observed in WL and PC chickens, respectively, and these differences in the expression pattern of genes analyzed may contribute to the pathogenicity of MD in each chicken line. 

This manuscript provides useful information for understanding the susceptibility and pathogenesis of MD. I read it with great interest. I also feel that each molecule and its potential impact on pathogenesis is well discussed. However, there is a part in the text where it is unclear whether the change in gene expression is WL or PC, and I think it would be easier to understand if you could correct that part. Also, please make the images in the figures clearer and clarify which group was compared for the statistical tests.

Minor comments

- Objectives: in addition to PRs, the authors analyzed other factors that has the potential involved in the pathogenesis (non-PR regulatory gene). Therefore, the objectives of this study should be modified in accordance with the actual analyses.

- Methods: are there any differences in virus dissemination between WL and PC shedders?

- L105: this sentence should be deleted.

- L217: please clarify the comparison of the letters indicated by “a” to “e”.

- L224-225: is this a description about WL and PC chickens exposed to PC chickens ? Also, are these compared with age-matched control chickens? Please clarify.

- L275-276: according to Figure 3e, the expression levels of stat5a in PC chickens seems to be higher than those in WL chickens. Please confirm.

- L284: hemostasis? homeostasis? Please confirm.

- L291-293: are these describe

Comments on the Quality of English Language

I don't feel any major problems in English.

Author Response

We thank the reviewers for their time spent reading our manuscript and for their helpful comments. These comments have strengthened our work considerably, and we hope to have addressed all concerns to your satisfaction.

Reviewer 1

In the present study, the authors analyzed the expression patterns of PR genes in MD-resistant and -susceptible chickens in lungs at early infection. In addition, the authors compared the expression levels of candidate molecules involved in the pathogenesis between the groups. The altered expression was observed in WL and PC chickens, respectively, and these differences in the expression pattern of genes analyzed may contribute to the pathogenicity of MD in each chicken line. 

This manuscript provides useful information for understanding the susceptibility and pathogenesis of MD. I read it with great interest. I also feel that each molecule and its potential impact on pathogenesis is well discussed. However, there is a part in the text where it is unclear whether the change in gene expression is WL or PC, and I think it would be easier to understand if you could correct that part. Also, please make the images in the figures clearer and clarify which group was compared for the statistical tests.

Response: Thank you for this comment. It is an important point made by the reviewer. The following information has been added:

  • In the figure legends, the following is added. “For specific genes, groups with different superscripts (a-d) are significantly different(P < 0.05) to each other.’
  • Added to 2.6. Statistical Analysis section (Lines 189-191)

The comparisons were made between the groups, including age-matched control and exposed groups. Superscripts (a-e) were used to indicate significant differences between the groups.”

 Also, we have modified the figures to make them clearer.

Minor comments

- Objectives: in addition to PRs, the authors analyzed other factors that has the potential involved in the pathogenesis (non-PR regulatory gene). Therefore, the objectives of this study should be modified in accordance with the actual analyses.

Response: Thank you for this comment. We have included the statement, “In addition, select non-purinergic PR regulatory gene responses were measured” in the abstract (Lines 15-16)

- Methods: are there any differences in virus dissemination between WL and PC shedders?

Response: This is a good question. Though we do not have quantitative data to support this, we suspect the MD-susceptible (PC) shed more virus than the MD-resistant (WL) chickens. Thus, to address the potential differences when performing the exposure experiments, we used both chicken lines. The dose of virus chickens were exposed to would likely affect responses and may explain the typically greater responses when chickens were exposed to the PC chickens.

- L105: this sentence should be deleted.

Response: We deleted this.

- L217: please clarify the comparison of the letters indicated by “a” to “e”.

Response: For clarification, more information has been added to the 2.6. Statistical Analysis section (L189-191), as well as in Figures 2-6 legends.

- L224-225: is this a description about WL and PC chickens exposed to PC chickens? Also, are these compared with age-matched control chickens? Please clarify.

Response: Thank you for noticing this. The text has been updated to clarify the information. “We observed a significant increase in DDX5 expression at 6 h in MDV-infected WL and PC chickens exposed to PC birds compared with age-matched controls. The increase was even more pronounced in PC chickens at 24 h while back to basal levels for WL chickens (Figure 3a).” Lines 227-230

- L275-276: according to Figure 3e, the expression levels of stat5a in PC chickens seems to be higher than those in WL chickens. Please confirm.

Response: Although the levels of STAT5a in PC chickens are higher than in WL chickens, the differences were not statistically significant as both are annotated with the same number for each group.

While re-examining the data, we found that the paragraph was unclear, and it has been revised to reflect this (L279-283).

- L284: hemostasis? homeostasis? Please confirm.

Response. Thank you for catching this mistake. It should be homeostasis and has been corrected (L291).

- L291-293: are these describe

Response: It is not clear what the reviewer is asking. We presume they want to know if the role of SOD and CAT in oxidative stress have been described.  We had references for this, assuming that is what the reviewer requested.

Reviewer 2 Report

Comments and Suggestions for Authors

The present study conducted gene expression analyses of purinergic receptors in PC and WL chickens horizontally exposed to MDV. Overall the manuscript is well written and results well presented.

Minor concerns.

1. Is there any polymorphic variation in the examined genes among PC and WL chickens?

2. Did the authors investigate PR agonist/antagonist effects on MDV replication/latency in vitro?

Author Response

Responses to reviewers' comments are in bold.

We thank the reviewers for their time spent reading our manuscript and for their helpful comments. These comments have strengthened our work considerably, and we hope to have addressed all concerns to your satisfaction.

Reviewer 2

The present study conducted gene expression analyses of purinergic receptors in PC and WL chickens horizontally exposed to MDV. Overall the manuscript is well written and results well presented.

Minor concerns.

  • Is there any polymorphic variation in the examined genes among PC and WL chickens?

Response: This is a great question.  We do not know whether there are polymorphic variations in the genes examined. The primers designed had been previously published or designed in our laboratory based on GeneBank sequences (Table 1). A majority of the primers designed worked in our assays, while a select few (not included in the study) did not amplify sequences, which could be related to genetic differences in primers binding. This would be a good area to address in the future comparing the different chicken lines.

  • Did the authors investigate PR agonist/antagonist effects on MDV replication/latency in vitro?

Response: Thank you for highlighting this important point. We have not performed PR agonist/antagonist effects during MDV replication thus far. One issue we have is that the tissue culture system used for in vitro infection is much different from the natural infection, and thus, we it would be difficult to fully address specific PR responses in vivo using the cell culture system.